# Guideline Adherence of Paediatric Urolithiasis: An EAU Members’ Survey and Expert Panel Roundtable Discussion

**DOI:** 10.3390/children9040504

**Published:** 2022-04-02

**Authors:** Beatriz Bañuelos Marco, Bernhard Haid, Anna Radford, Thomas Knoll, Sajid Sultan, Anne-Françoise Spinoit, Manuela Hiess, Simone Sforza, Rianne J. M. Lammers, Lisette Aimée ‘t Hoen, Edoardo Bindi, Fardod O’Kelly, Mesrur Selçuk Silay

**Affiliations:** 1Department of Urology and Paediatric Urology, Charité Medical University of Berlin, 10117 Berlin, Germany; 2Department of Paediatric Urology, Ordensklinikum Linz Hospital of the Sisters of Charity Linz, 4010 Linz, Austria; bernhard.haid@me.com (B.H.); manuh@gmx.at (M.H.); 3Department of Paediatric Urology, Leeds Teaching Hosptials NHS Trust, Leeds LS1 3EX, UK; anna.radford@nhs.net; 4Department of Paediatric Surgery, Hull University Teaching Hospitals NHS Trust, Hull HU3 2JZ, UK; 5Hull York Medical School, University of York, Heslington, York YO10 5DD, UK; 6Department of Urology, Klinikum Sindelfingen-Boeblingen, University of Tuebingen, 71065 Tuebingen, Germany; t.knoll@klinikverbund-suedwest.de; 7Department of Paediatric Urology, Sindh Institute of Urology and Transplantation, Karachi V266+CR, Pakistan; sajidsultan818@hotmail.com; 8Department Urology ERN Centre, Ghent University Hospital, Ghent University, 9000 Ghent, Belgium; afspinoit@hotmail.com; 9Paediatric Urology, Meyer Children Hospital, University of Florence, 50121 Florence, Italy; simone.sforza1988@gmail.com; 10Department of Urology, University Medical Center Groningen, 9713 GZ Groningen, The Netherlands; rianne_lammers@hotmail.com; 11Department of Paediatric Urology, Sophia Children’s Hospital, Erasmus University Medical Center, 3015 GD Rotterdam, The Netherlands; l.thoen@erasmusmc.nl; 12Ospedale Pediatrico G Salesi Paediatric Urology, Departments of Urology and Pediatric Urology, G Salesi Paediatric Hospital, 00165 Rome, Italy; edo.bindi88@hotmail.it; 13Division of Paediatric Urology, Beacon Hospital, DK18 AK68 Dublin, Ireland; fokelly@rcsi.ie; 14School of Medicine, University College Dublin, D04 V1W8 Dublin, Ireland; 15Department of Urology, UPMC Kildare, W91 W535 Clane, Ireland; 16Division of Pediatric Urology, Department of Urology, Biruni University, 34010 Istanbul, Turkey; selcuksilay@gmail.com

**Keywords:** paediatric nephrolithiasis, paediatric radiology protection, percutaneous, nephrolithotomy, shockwave lithotripsy, micro-PNL, mini-PNL, stone recurrence, residual fragments

## Abstract

Background: Paediatric nephrolithiasis has increased globally, requiring standardized recommendations. This study aims to assess the paediatric urolithiasis care between EAU members along with the statements of three experts in this field. Methods: The results of an electronic survey among EAU members comparing the guideline recommendations to their current practice managing paediatric nephrolithiasis in 74 centres are contrasted with insights from an expert-panel. The survey consisted of 20 questions in four main sections: demographics, instrument availability, surgical preferences and follow-up preferences. Experts were asked to give insights on the same topics. Results: A total of 74 responses were received. Computerised Tomography was predominantly used as the main imaging modality over ultrasound. Lack of gonadal protection during operations was identified as an issue. Adult instruments were used frequently instead of paediatric instruments. Stone and metabolic analysis were performed by 83% and 63% of the respondents respectively. Conclusions: Percutaneous Nephrolithotomy is the recommended standard treatment for stones > 20 mm, 12% of respondents were still performing shockwave lithotripsy despite PNL, mini and micro-PNL being available. Children have a high risk for recurrence yet stone and metabolic analysis was not performed in all patients. Expert recommendations may guide clinicians towards best practice.

## 1. Introduction

The incidence of urolithiasis in paediatric populations varies from 1–5% to 5–15% in advanced and developing countries respectively although globally it appears to be increasing [1,2,3]. The composition of urine in the paediatric population contains increased citrate and magnesium, associated with inhibition of crystal formation [4]. Some recent literature demonstrates that, in emerging countries, the surgical burden of stone disease grew by 62% between 1998 and 2015 [5].

The European Association of Urology (EAU) produced and continually updates guidelines to standardize recommendations on diagnosis and treatment based on the best available evidence [6].

Shockwave lithotripsy (SWL) is the prevailing recommendation for paediatric urolithiasis as it is the least invasive approach. The recommendations of the EAU/ESPU are similar to those of adults [7] (see Appendix A, Table A1). However, the literature providing the basis of these recommendations consists mainly of non-randomized retrospective studies resulting in a level of evidence (LE) of 2 for almost all treatment recommendations.

Our aim was to assess the care of paediatric urolithiasis. We report the results of an electronic survey among EAU members comparing EAU guideline recommendations to their own current practice managing paediatric urolithiasis, along with the statements of three eminent experts in this field. 

## 2. Materials and Methods

An electronic survey containing twenty questions, available through Survey Monkey, was carried out and the link highlighted in the e-bulletin was sent to all members of the EAU. The questions addressed the following key areas: demographics of respondents and centres, treatment modalities, imaging techniques, equipment utilization, metabolic/stone composition assessment and evaluation rates. A comparison with EAU guidelines was made. Only fully completed questionnaires were included in the final analysis.

The experts were identified by performing a literature review and a group discussion within the EAU Young Academic Urologists, paediatric urology expert group. They were contacted via e-mail and asked to respond in written form to the questions attached to an invitation letter.

We feel that the topic we addressed in this article is relevant for these patients and the clinicians treating them.

The expert panel included: −SS: Professor Sajid Sultan, Head of the Department, Paediatric Urology at Sindh Institute of Urology and Transplantation, Kirachi, Pakistan;−MSS: Professor Mesrur Selçuk Silay, Paediatric Urologist Istanbul Biruni, University Director/Paediatric Robotic Surgery Programme, Memorial Hospital Group;−TK: Professor Thomas Knoll Associate Professor of Urology at Mannheim University Hospital, Germany, Head of the Department of Urology, Sindelfingen Medical Centre, University of Tubingen.

## 3. Results

### 3.1. Electronic Survey Results

Overall, 218 members responded from 87 countries. Complete responses were only obtained from 33.9% (*n* = 74). In Table 1, the results of the survey are described: demographics of respondents and centres. 

SWL was available as a treatment modality in 65/74 centres (87.8%). Ureteroscopy was available as either flexible 79.7% (59/74) or semirigid 94.59% (70/74), and mini URS was defined as 4.5Fr (12) 24% (32/74). Standard PNL was available in 91.89% (70/74) of centres, mini PNL in 66.22% (49/74), and micro PNL in 13/74 centres (17.6%)

The majority of centres (75.7%; 56/74) performed ≤ 10 PNL procedures annually, with only 5.4% centres performing >30 PNL procedures per year. As expected, Amplatz sheath size was directly correlated with patient age.

Stones less than 1 cm in size (irrespective of their localization) would—according to respondents—be treated initially using SWL. With an increase in theoretical stone size from 1 cm to 2 cm, (63.5%, *n* = 47) respondents would still choose to use SWL. In stones > 2 cm, the initial preference still demonstrated significant variability with 12.2% (*n* = 9) opting to utilize SWL, but with the majority (71.6%, *n* = 53) preferring PNL (Table 2). 

Prior to PNL (Percutaneous Nephrolithotomy)/fURS, (Flexible Ureterorenoscopy), the CT (Computerized Tomography) scan was the preferable imaging method (56.8% = 42/74) with 37/74 (50%) choosing to additionally perform ultrasound (US). Only 42% of centres utilizing PNL/fURS protect the gonads during the screening procedure, with 32.4% reporting that protection was used occasionally, and a further 25.7% reporting that they did not use gonadal protection at all (Table 3).

### 3.2. Expert Panel Questions

The 2020 EAU Guidelines recommend a complete metabolic evaluation as urinary metabolic studies are infrequently, and in many cases, not adequately, performed.

All three experts agreed with this recommendation. This should ideally be carried out with a 24 h urinary metabolic quantitative study (urine uric acid, calcium, citrate, oxalate, magnesium, potassium, sodium, proteins and qualitative tests for cysteine). A biochemical stone analysis must be performed, being relevant for individual risk stratification. MSS and TK agreed in performing the metabolic study around 4 weeks after surgery in order to elucidate the composition of the stone, as indicated in the EAU guidelines [8]. 

### 3.3. Diagnostic Imaging

Non-complicated stone cases (1–2 stones <10 mm; No anatomic abnormality): All three agreed that US is sufficient, also as a follow up. The location and the size of the stone can play an important role in this choice. If visibility in the US is compromised or it shows abnormalities, further studies should be performed. 

Complicated stone cases (>2 stones, >10 mm, staghorn stones, bilateral stones): MSS and TK stated that non-contrast CT could be useful whereas SS requires it only for staghorn calculus or multiple non opaque stones. 

Concomitant anatomical abnormalities (obstruction of uretero-pelvic junction [UPJO], obstruction of uretero-vesical junction [UVJO], Pelvic kidney, horseshoe kidney, solitary kidney): All three experts find CT scans useful for providing details regarding vascularity, stone complexity, and detailed anatomy. 

In the case of a spontaneous passage of the stone and no stone analysis: All three experts agreed on US for surveillance imaging, carrying out blood tests and 24 h urine analysis.

### 3.4. Stone Instruments

Technique availability: None of the experts have the problem of availability. According to MSS, a stone institution dealing with children should ideally have minimum equipment readily available.

8F paediatric compact cystoscope;4.5F ultrathin semirigid ureteroscope;4.9-7.5F flexible ureteroscope (either digital or conventional);Mini-PNL instrument allowing performance of PNL through 13F- to 20F amplatz sheath;Holmium laser with minimum 30 W energy;In addition, microperc-4.8F, ultraminiperc-13-14F or superminiperc-13F instruments, high power laser technology, ultrasonic fragmentation, various types of basket catheters.

### 3.5. Techniques (as Available)

SWL: Ureteral stones, pelvic, mid- and upper pole stones < 10mm where complete clearance in one or two sessions is highly probable. Dense stones with >1200 Hounsfield units (HU) and cysteine stones are not subjected to SWL.

RIRS (Retrograde intrarenal surgery): MSS declares using this technique for pelvic, mid- and upper-pole stones between 10 and 20 mm. SS recommended it for upper ureteric and renal stones up to 2 cm, though it is rarely performed because of the necessity of a ureteric access sheath (UAS) being the minimum size available, 9Fr, which is too big for small children. TK generally only performs it in children >6 years of age. 

Standard PNL. All the experts avoid it.

Mini PNL: Used by all the three experts generally for > 20 mm and lower pole stones between 10 and 20 mm, when SWL fails or cannot be applied or when RIRS is not desirable.

MSS highlights the importance of validated nomograms for all lower pole stones <10 mm that predict the success of SWL for clinical judgement [9]. When the likelihood of the success of SWL is not high, then PNL or RIRS might be used. 

For MSS and SS, the miniaturization of PNL affected their clinical practice; TK had already modified the technique using a comparatively sized ureteroscope. Each of them also agreed on using miniperc with Amplatz 14Fr, 14–16Fr for pre-school children using a semi-rigid nephroscope (10–12Fr), whereas 14–20Fr for older children. Micro PNL offers a direct puncture for isolated single calyceal stones < 15 mm or in calyces with a narrow infundibulum.

### 3.6. Approach in Anomalous Kidneys (Pelvic; Horseshoe)

SS uses PCNL in Horseshoe kidneys for the majority of the stones, preferably through posterior superior calyx, in pelvic kidney open surgery or laparoscopic assisted PNL. 

MSS—if accessible by RIRS, it is the first choice. If not, laparoscopy assisted procedures (miniperc, microperc or antegrade RIRS) through one of the trocars; previously published [10].

TK—depending on stone location and anatomy, PNL and RIRS would probably be the preferable techniques.

### 3.7. Concomitant Urological Anomalies (UPJO; UVJO)

All three experts agreed on correcting anomalies with the removal of the stone during the same surgery if technically possible. The surgical approach to these cases varies. SS prefers using open surgery and laparoscopy for stones in UPJO in specific cases. MSS prefers minimally invasive surgery, either laparoscopic or robotic in combination with antegrade RIRS or PNL. TK prefers UPJO robotic minimally invasive surgery.

## 4. Discussion

The survey data from 78 centres alongside the opinions from three experts in the field highlight many controversial points concerning the diagnosis and management of nephrolithiasis in the paediatric population, which were often found to diverge from the EAU guidelines. This was especially apparent in relation to metabolic evaluation, but also regarding the use and availability of equipment adapted for use in paediatric populations. More evidence is clearly required for a uniform and targeted treatment. All three experts agreed that the current EAU guidelines are a useful instrument for clinical decision-making. There were some limitations that were stated, such as the necessity for individual treatment plans based on stone composition (HU), the availability of appropriate facilities and local expertise, cost effectiveness and variable socioeconomic conditions, patient preference, and the non-inclusion of paediatric non-randomized controlled trials or large case series which may also play a role, and have value, in clinical decision making.

As low as reasonably achievable (ALARA) principles for radiation exposure are an integral part of the EAU guidelines [7,11,12], as cumulative radiation has been proven to be a risk for developing malignancies later in life (13). CT scanning imparts a lifetime risk of cancer if performed during childhood, with one CT amounting up to the radiation burden of 10 KUBs, dependent on local protocol (5–8 vs. 0.5 mSv) [13]. The increasing prevalence of low-dose CT use in children aims to reduce cumulative radiation burden without decreasing image quality [14]. 

US is strongly advocated for the detection and management of stone disease in the paediatric population [15]. According to the EAU guidelines, abdominal CT should only be required in cases with diagnostic uncertainty, or high complexity (LE:2, GR B). Although there is a need for continued research, newer studies have highlighted the limitations of US in ureteric calculi, and the diagnostic potential of unenhanced CT due to its higher sensitivity [16]. The experts confirmed US as the modality of choice, choosing CT only in complicated scenarios, or significant urological anomalies. Interestingly, only 50% of EAU member respondents used US as the pre-operative radiological method of choice, and 62.2% post-operatively. Only 41.9% of respondents utilized gonadal shields at all times. As the developing gonads are particularly sensitive to radiation in the paediatric population, these findings are unexpected and concerning; however, there may be some response bias in this survey, and the use of gonadal protection measures may be a regional issue. The experience (case volume) of each centre in managing urolithiasis is also likely an important factor that may influence the use of different imaging techniques and gonadal protection. 

In adults, there is a trend to use progressively smaller Amplatz sheaths for PNL, and therefore it might be expected that for children, smaller sheaths down to 12Ch/Fr would be used [17]. Tubeless PNL in well-selected cases has been performed and is safe [18]. When the experts were asked about PNL, none of them used the standard technique when operating on children, and they agreed on using Miniperc 14–16Fr Amplatz sheaths for pre-school children with a semi rigid nephroscope (10–12Fr), whereas 14–20Fr was used for older children. Thus, it is surprising that the survey results showing that most respondents (64%) used 16Ch in the 0–2 age group. As described in the results section, the majority of centres (75.7%; 56/74) performed ≤10 PNL procedures annually, with only 5.4% centres performing >30 PNL procedures per year. This low volume may reflect the lack of paediatric expertise and may be a reason why the specific equipment is being utilized in this manner across different age cohorts. We agree with the required paediatric urology equipment stated by MSS, as it is been proven that larger sheath size is statistically associated with more postoperative complications [19]. It appears that the lack of specialized equipment availability, and inexperience with paediatric urolithiasis can lead to the use of adult instruments [20,21].

SWL is the recommended treatment for stones < 2.0 cm as it is the least invasive. Success rates of SWL are relatively high compared (Stone Free Rate 57%–92% in long term) to the adult population and fragments after SWL treatment are easier and more quickly eliminated by children [22]. Re-treatment rates are quoted from 13.9% to 55.9%, with further subsequent interventions required in 7%–33% patients. Of the respondents, 87.8% (*n* = 65) utilized SWL for the treatment of stones < 1 cm, in line with EAU guidelines, correlating with availability (65/74 respondent centres). This renders approaches, such as PNL, which is more likely to yield complete stone clearance in one session, a more attractive option for experienced urologists even with small stones [17]. This is further highlighted, considering the evolution to mini and micro PNL proving optimal safety with fewer complications and high efficiency with low residual fragment rates [5,12,18].

Despite recommendations to use PNL for stones > 20 mm, 12% of respondents still utilized SWL to treat these stones, even in centres where this technique was available. This might be associated with the low number of PNL performed by the majority of the centres. One of the main problems of SWL is its putative effect on kidney maturity [23]. 

PNL on the other hand achieves stones clearance between 71.1%–97.3% at after the first session, and 84.6%–97.5% after a second look [18,24,25] with an overall complication rate of 20% related to both the size of the nephroscope and stone [26]. After DMSA scan evaluation, the rates of focal renal damage using PNL tracts between 12 and 24Ch was 5% [27]. RIRS is also gaining popularity in paediatric stones disease treatment, especially for stones between 10–20 mm. Our experts and our survey echoed this strategy, however RIRS may require multiple general anaesthetic procedures, can be technically challenging, and imparts a significant institutional cost [28]. 

Because paediatric urolithiasis is often as a result of underlying metabolic, anatomic or functional disorders, recurrences are common [22]. EAU guidelines stipulate that a complete metabolic screen should be performed based on stone analysis, which requires the collection of stone material to be analysed. Our survey shows that 83.8% of respondents perform stone analysis, and only 66.2% perform a metabolic evaluation. The importance of a proper metabolic analysis, 24 h urinalysis and the relevance of stone composition were emphasized and endorsed by all three experts, including how to manage clinically insignificant residual fragments (CIRF).

Although there are no clear recommendations as to how to define CIRF in children, many authors agree that children should be completely stone free, as approximately one third of patients will progress to develop clinically significant residual fragments (RF) [29]. SWL might result in RF more often; however, it is clearly less invasive for children compared with PNL. 

All three experts agreed that children should be completely stone free, however as stated by SS, small RF are not uncommon especially post SWL. In his experience, RFs of any size from 3 to 6 mm may pass spontaneously; however, as the fragment size increases from 4 to 6 mm, the chances of stone re-growth increases and clearance decreases. Patients with RFs of any size after SWL require close follow up and timely intervention if needed. In his series of 94 RFs post SWL, 50% of those equal to or less than 3 mm passed spontaneously compared to 15% of those 6 mm in size. Furthermore, 7/94 required surgical intervention due to symptoms, all of whom had RFs from 4–6 mm [5]. MSS also highlighted the importance of the size of the RF and the composition of the stone. Cystine stones tend to recur quickly, and infection-associated stones should completely be eliminated without any RF. For calcium stones, spontaneous passage can be contemplated for RF less than 3 mm. During this follow-up, potassium citrate treatment may also be offered. Another observation by MSS is the evaluation of the RF postoperatively, there should be a minimum of a 3-month interval after the procedure. Earlier diagnosis may lead to unnecessary further procedures.

## 5. Limitations

Possible limitations of this study include the relatively low number of respondents as well as the relatively low rate of completed questionnaires (33.9%, *n* = 74). As the design of the form was reported as user-friendly, brief, and there were no technical concerns, there is no clear explanation for this. Nevertheless, each of the 74 respondents reflects the practice of an individual centre, which we feel helps validate and amplify the impact of our results, especially bearing in mind the geographic distribution of the respondents. It is also a snapshot of how paediatric urolithiasis is being managed regardless of EAU guideline recommendations, possibly as a result of a lack of expertise and/or instrument availability. The number of cases performed in the responding centres could add another limitation to the study as many of the respondents were not in high-volume centres. 

The expert panel questions were conceived using a Delphi consensus method by the Paediatric Urology Expert Group of the YAU, which we concede may be open to bias. However, we feel that the topics addressed in this article are timely and important for specialised patient management. 

## 6. Conclusions

Contrary to guideline recommendations that SWL be used for stones <20 mm and not for those >20 mm, this survey also demonstrates its utilization for larger stones. Imaging often does not conform to ALARA principles, or comply with EAU guideline recommendations and warnings for the use of CT scanning. Gonadal protection is not routinely used.

The increasing use of PNL in smaller stones may reflect the evolution of this technique and its results in reducing stone burden.

Our findings reflect the lack of available data in the literature and could signify a more important role for SWL in larger stones, as well as PNL in smaller stones. In the absence of evidence for these widely used protocols, this should prompt the initiation of carefully designed prospective clinical evaluations. The indications and safety of unenhanced CT should be evaluated.

The results of the survey showed a relatively low compliance towards the EAU Guidelines on paediatric stone disease in those responding centres. This, however, could be slightly misleading given the differences in in volume across different stone centres. If this is an issue of volume and technique availability, then the obvious question posed is whether paediatric stone management should be centralized in order to deliver high-quality evidence-based patient care.

## Figures and Tables

**Table 1 children-09-00504-t001:** Demographics of the respondents for the Survey on paediatric urolithiasis care from higher to lower percent in each area (Age, institution and country).

Survey Respondents Demographics		
Response per Age Bracket	26–35 years	34.8%
	36–45 years	31.6%
	46–55 years	17.8%
	56–65 years	12.4%
Academic Institution	66.1%	
Non Academic Institution	22.5%	
Private Hospital	11.4%	
Countries with Higher Rate of Response	Turkey	14.7%
	Germany	11.9%
	UK	10.9%
	Italy	6.9%
	Spain	5.5%
	Poland	5%
	Portugal	4.6%

**Table 2 children-09-00504-t002:** Treatment modalities, utilization in the EAU centres responding to the survey depending on the size of the stone and used size of Amplatz Sheath depending on age of the patient.

Treatment Modalities and Equipment Utilization by the Surveyed Centres							
Treatment Modalities and Equipment Utilization from higher to lower rate.							
Technique	Semi-Rigid URS	Standard PNL	SWL	fURS	Mini PNL	Mini URS 4.5Fr	Micro PNL
Availability between 74 centers N (%)	70 (94.5%)	70 (91.8%)	65 (87.8%)	59 (79.7%)	49 (66.2%)	32 (24%)	13 (17.6%)
Treatment Modalities and Equipment Utilization Depending on the Size of Stone.							
Size of Stone (cm)	SWL		fURS		PNL		
<1 cm	65 (87.8%)		10.8% (*n* = 8)		1.4% (*n* = 1)		
1–2 cm	63.5%, *n* = 47		23.0% (*n* = 17)		13.5% (*n* = 10)		
>2 cm	12.2% (*n* = 9)				(71.6%, *n* = 53)		
Use of Amplatz Sheath according to Size and Age of the Patient.							
Use of Amplatz Sheath 16Ch			<2 years				
			64%				
Use of Amplatz Sheath 16Ch			13–18 years				
			12%				
Use of Amplatz Sheath 30Ch			13–18 years				
			6%				
Use of Amplatz Sheath 30Ch			<2 years				
			No use				

SWL: Shockwave lithotripsy. fURS: Flexible Ureterorenoscopy. URS: Ureterorenoscopy. PNL: Percutaneous Nephrolithotomy. KUB: Kidney ureter bladder radiography. IVP: Intravenous Pielography.

**Table 3 children-09-00504-t003:** Metabolic and stone assessment, gonad protection and image modality prior and post PNL or fURS as performed in the surveyed centres.

Metabolic and Stone Assessment, Gonad Protection and Preferred Image Modality			
Stone Analysis	62/74 (83.8%)	Gonad Protection During PNL/URS	Routinely 42%
Metabolic Screen	49/74 (66.2%)		Occasionally 32.4%
No Stone or Metabolic Assessment.	1/74 (1.3%)		None 25.7%
Image Modality Prior to PNL/fURS		Image Modality post PNL/fURS	
CT *n* = 42 (56.8%)		US 46 (62.2%)	
US *n* = 37(50%)		KUB (47.2%)	
KUB/IVP *n* = 43 (58.1%)		CT *n* = 10 (13%)	

## Data Availability

Not applicable.

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
