# Peer review of "Guideline Adherence of Paediatric Urolithiasis: An EAU Members’ Survey and Expert Panel Roundtable Discussion"

_children, 2022, doi:10.3390/children9040504_

Round 1

Reviewer 1 Report

The authors present "Guideline Adherence of Paediatric Urolithiasis: An EAU  Members’ Survey and Expert Panel Roundtable Discussion" . This is an interesting manuscript, which could be improved after revision.

 Comments:

Abstract section

-Abstract overall provides a fair summary of the article

-"Conclusions: Despite Percutaneous Nephrolithotomy is the recommended standard treatment for stones >20mm, 12% of respondents were still performing Shockwave lithotripsy despite PNL, mini and micro-PNL being available". Please rephrase it as the double use of "despite" confuses the readers

Introduction section
-The introduction is clear, useful in setting the scene and  understandable but there are  few, very minor, errors in sentence construction

Materials and methods section 

-"We feel that the topic we addressed in this article is relevant for these patients and the clinicians treating them" This sentence is not applicable  in this section of the study

Results section

-"Table 1. Demographics of the respondants for the Survey on paediatric urolithiasis care and  guideline adherence between EAU members".

I cannot see any data on guideline adherence between EAU members in table 1.

Please present response rate from higher to lower percentages. Generally response rate was low. How this may influence results. Please discuss on this in discussion section

-Table 2 presents data on different questions and it is confusing . I suggest to be spited in 2 tables

 Discussion section

Please discuss the results not only regarding the availability of equipment but also considering experience of centres  in pediatric urology (ex. number of stone cases per year)

Author Response

Dear reviewers,

Thank you very much for your comments and insightful remarks. We have tried to respond as clearly as possible to your comments/suggestions and all the changes demanded in the manuscript have been performed, including an addition of limitations section in the discussion.

We hope we have achieved a better outcome and we thank again for contributing to make the manuscript clearer and more accurate in our presentation of this data. We are grateful for the time the reviewers invested in their comments and suggestions.

Reviewer 1

Abstract section

-Abstract overall provides a fair summary of the article

-"ConclusionsDespite Percutaneous Nephrolithotomy is the recommended standard treatment for stones >20mm, 12% of respondents were still performing Shockwave lithotripsy despite PNL, mini and micro-PNL being available". Please rephrase it as the double use of "despite" confuses the readers

Thank you very much for the observation, it has been corrected accordingly.

Introduction section
-The introduction is clear, useful in setting the scene and  understandable but there are  few, very minor, errors in sentence construction

Thank you very much, we have taken care to revise it once again. We hope it has been improved sufficiently.

Materials and methods section 

-"We feel that the topic we addressed in this article is relevant for these patients and the clinicians treating them" This sentence is not applicable  in this section of the study

Thank you very much for the commentary, we have moved the sentence to the introduction section.

Results section

-"Table 1. Demographics of the respondants for the Survey on paediatric urolithiasis care and guideline adherence between EAU members".

I cannot see any data on guideline adherence between EAU members in table 1.

Thank you very much for the correction, we have changed the description of the table accordingly.

Please present response rate from higher to lower percentages. Generally response rate was low. How this may influence results. Please discuss on this in discussion section

Thank you very much for the correction, we have changed the description of the table accordingly. The low response rate is low and certainly affecting the results, therefore we have taken your comment and addressed it in the discussion.

-Table 2 presents data on different questions and it is confusing. I suggest to be spited in 2 tables

Thank you very much for your suggestion, we believe it is clearer after splitting it in 2 and making other modifications to make it visually understandable.

 Discussion section

Please discuss the results not only regarding the availability of equipment but also considering experience of centres in pediatric urology (ex. number of stone cases per year)

Thank you very much for your comment; this is absolutely right and has been included in the discussion and in a limitations section as well.

Reviewer 2 Report

The authors conduct a survey of clinical practice of different centers in Europe. I find it interesting and honest as a survey of still rigid approach to the subject of treating urolithiasis in children. However, I would like to see a possible regional differences in practice according to GDP without the mention of the particular Countries . Otherwise I have no objections

Author Response

Dear reviewers,

Thank you very much for your comments and insightful remarks. We have tried to respond as clearly as possible to your comments/suggestions and all the changes demanded in the manuscript have been performed, including an addition of limitations section in the discussion.

We hope we have achieved a better outcome and we thank again for contributing to make the manuscript clearer and more accurate in our presentation of this data. We are grateful for the time the reviewers invested in their comments and suggestions.

Reviewer 2

The authors conduct a survey of clinical practice of different centers in Europe. I find it interesting and honest as a survey of still rigid approach to the subject of treating urolithiasis in children. However, I would like to see a possible regional differences in practice according to GDP without the mention of the particular Countries . Otherwise I have no objections.

Thank you very much for your valuation of our work and the mention of the possible regional differences. We have add this as a comment in the discussion section, as this was an electronic survey some of the data we can´t access anymore unfortunately which impedes us for being able to confirm if there are possible regional difference in practice, but it seems reasonable.

Round 2

Reviewer 1 Report

I have no further comments